# Cumulative Effect Assessment of Common Genetic Variants on Prostate Cancer: Preliminary Studies

**DOI:** 10.3390/biomedicines10112733

**Published:** 2022-10-28

**Authors:** Anca Gabriela Pavel, Danae Stambouli, Gabriela Anton, Ismail Gener, Adrian Preda, Catalin Baston, Constantin Gingu

**Affiliations:** 1Cytogenomic Medical Laboratory, Molecular Genetics Department, 014453 Bucharest, Romania; 2The Romania Academy, “Stefan S. Nicolau” Institute of Virology, 030304 Bucharest, Romania; 3Department of Nephrology, Urology, Immunology and Immunology of Transplant, Dermatology, Allergology, Faculty of Medicine, “Carol Davila” University of Medicine and Pharmacy, 010751 Bucharest, Romania; 4Department of Nephrology, Fundeni Clinical Institute, 022328 Bucharest, Romania; 5Center of Urological Surgery, Dialysis and Renal Transplantation, Fundeni Clinical Institute, 022328 Bucharest, Romania

**Keywords:** polymorphism, prostate cancer, cumulative effect, genetic risk, genotyping

## Abstract

Single nucleotide polymorphisms (SNPs) are the most common type of genetic variation among people. Genome Wide Association studies (GWASs) have generated multiple genetic variants associated with prostate cancer (PC) risk. Taking into account previously identified genetic susceptibility variants, the purpose of our study was to determine the cumulative association between four common SNPs and the overall PC risk. A total of 78 specimens from both PC and benign prostate hyperplasia (BPH) patients were included in the study. Genotyping of all selected SNPs was performed using the TaqMan assay. The association between each SNP and the PC risk was assessed individually and collectively. Analysis of the association between individual SNPs and PC risk revealed that only the rs4054823 polymorphism was significantly associated with PC, and not with BPH (*p* < 0.001). Statistical analysis also showed that the heterozygous genotype of the rs2735839 polymorphism is more common within the BPH group than in the PC group (*p* = 0.042). The cumulative effect of high-risk alleles on PC was analyzed using a logistic regression model. As a result, the carriers of at least one risk allele copy in each particular region had a cumulative odd ratio (OR) of 1.42 times, compared to subjects who did not have any of these factors. In addition, the combination of these four genetic variants increased the overall risk of PC by 52%. Our study provides further evidence of the cumulative effects of genetic risk factors on overall PC risk. These results should encourage future research to explain the interactions between known susceptibility variants and their contribution to the development and progression of PC disease.

## 1. Introduction

Prostate cancer (PC) is the second-most frequent cancer in men, with an incidence of 29.3/100,000 and a mortality rate of 7.6/100,000 worldwide [1]. Moreover, in Eastern Europe, even though the incidence rates of PC have continued to increase in recent years, the mortality rate has stabilized in many countries; these outcomes may reflect the new advancements in cancer treatment [2].

The most common method used to detect prostate conditions, including PC, benign prostatic hyperplasia (BPH), and prostatitis (inflammation of the prostate) is by measuring the serum levels of prostate-specific antigen (PSA) [3]. 

In addition to the known risk factors (increasing age, race, and family history) genetic variants or polymorphisms existing in the human genome are associated with a genetic susceptibility to cancer. Single nucleotide polymorphisms (SNPs) are the most common type of genetic variation among people, present in more than 1% of the population. There are millions of SNPs in the entire human genome; therefore, it is difficult to choose target SNPs that are most likely to be associated with the disease development. Genome Wide Association studies (GWASs) have yielded multiple SNPs associated with PC risk. Individually, these SNPs are moderately associated with the risk of PC [4]. However, when combined together, these SNPs have a stronger dose-dependent association [5,6]. This opens the possibility of using multiple SNPs for risk prediction.

The aim of our study was to find out if a combination of SNPs would have a stronger association with PC than each individual SNP in a group of Romanian patients with PC disease. In this regard, we have selected from the literature four SNPs that were previously found to be associated with PC overall risk in Western Europe and American populations: rs12422149 (11q13), rs4149117 (12p12), rs4054823 (17p12), and rs2735839 (19q13) [7,8,9,10,11,12]. Taking all four polymorphisms together, we calculate their cumulative effect on PC onset.

## 2. Materials and Methods

### 2.1. Study Subjects

A total of 78 male patients, of which 50 were diagnosed with PC and 28 with BPH, were included in this study. The selected patients were all Romanian Caucasians and were selected between 2014 and 2016 from a single center in Romania (Fundeni Clinical Institute, Bucharest). The samples used for DNA extraction were peripheral blood specimens collected in EDTA (ethylene diamine tetra acetic acid) tubes. The diagnosis of PC and BPH was established according to current international guidelines (EAU and AUA). The patients diagnosed with BPH, for which the clinical and laboratory evaluation did not suggest the presence of PC, were included in the control group. This study has been approved by the Ethical Committee of the Fundeni Clinical Institute in Bucharest, Romania (approval number: 43074/26.11.2014) and it was performed in accordance with the Declaration of Helsinki. Guidelines and regulations were followed for all methods. Samples were obtained with the informed consent of the participants prior to their inclusion in the study. Standard protocols were followed to ensure the confidentiality of personal data.

### 2.2. Follow-Up

The first post-treatment visit was scheduled one month after RP, followed by periodic visits at 3-month intervals for up to 2 years. The follow-up visit was focused on detecting the patients with biochemical recurrence (BCR) by measuring the PSA levels. BCR was defined as two consecutive rising PSA (≥0.2 ng/mL) levels obtained at least 3 months apart. The first detectable PSA level was defined as time to PSA failure. 

### 2.3. Genotyping 

A total of four SNPs, which are currently under debate as risk factors for PC, were investigated: rs2735839 (*KLK3-KLK2*, 19q13 intragenic region), rs12422149 (*SLCO2B1*, 11q13 cytogenetic location), rs4149117 (*SLCO1B3*, 12p12 cytogenetic location), and rs4054823 (17p12 chromosomal region). Genotyping of the selected SNP markers was performed using a predesigned TaqMan assay (Thermo Fisher Scientific, Carlsbad, CA, USA) as described previously [13]. Automated genotyping was performed using the StepOne v2.3 software (Applied Biosystems, Foster City, CA, USA), and the results were verified manually. 

### 2.4. Statistical Analysis

Patients’ clinical data were summarized as mean ± standard deviation (SD) for continuous variables, and as numbers and percentages for categorical variables. We examined associations of each SNP individually and in aggregate with PC risk. In order to assess the potential association between SNPs and PC, allele and genotype frequency were compared for both groups. The statistical significance of the association between SNP alleles, genotypes, and disease status was evaluated using the Fisher’s exact test. Logistic regression analysis was used to test the association between each of the four SNPs and clinicopathological parameters (nodal invasion, tumor stage, PSA serum levels, and Gleason score). We tested the cumulative effects of the four SNPs (rs4054823, rs4149117, rs12422149, and rs2735839) on prostate cancer by counting the number of risk alleles of each SNP in the logistic regression model. In order to measure relative risk (RR) of PC, the odds ratios (ORs) and their 95% confidence intervals (CIs) were estimated. In all analyses, a two-sided *p* values of less than 0.05 was considered statistically significant. The statistical analyses were performed using the GraphPad Prism 9 software. We performed post-hoc power analysis using the G*Power 3.1 software. The post-hoc analysis yielded more than 80% power with a calculated effect size of 0.86, given a significance level of 0.05 and sample sizes of 50 PC patients and 28 BPH patients.

## 3. Results

Clinical characteristics of the 78 participants are presented in Table 1 as mean ± standard deviation, or number of subjects (percentage). The calculated mean age for both groups was 65.4 for PC and 70.5 for BPH patients, respectively. Overall, patients diagnosed with PC were younger than those with BPH (*p* < 0.05). Positive surgical margins (PSM) were detected in four patients with PC, and only two of them had BCR following local treatment.

The frequency of all tested SNP genotypes in patients within each group is presented in Table 2. Statistically significant results regarding risk genotype distribution within the two groups were obtained for only two out of four analyzed SNPs. Therefore, from the analyzed data, we found that the heterozygous genotype of the rs2735839 (*KLK3-KLK2* intragenic region) polymorphism was the most frequent within the BPH group (*p* = 0.042). For the rs4054823 (17p12 intragenic region), the frequency of the CT and TT genotypes were significantly higher within the PC group compared to the BPH group (*p* < 0.001). Within the BPH group, the CC genotype had the highest frequency (43%). Regarding the two *SLCO* genes, for both rs12422149 (*SLCO2B1*) and rs4149117 (*SLCO1B3*), the most frequent genotype was the GG (74%—PC group; 61%—BPH group) and the TT genotype, respectively (74%—PC group; 72%—BPH group).

The risk allele frequency for all four SNPs was calculated. By evaluating the allele distribution within the two groups, we have identified a significantly higher carrier frequency of the T risk-allele of rs4054823 polymorphism within PC patients compared with the BPH group, with a 2.55-fold increased risk for PC (OR = 2.55, 95%CI: 1.44–4.52, *p* < 0.001). In this case, the OR is significantly different from 1, and, consequently, there is an increased RR in the PC group compared to the BPH group (Table 3).

In order to investigate whether the combined presence of particular gene polymorphisms affects the onset of PC, we examined the cumulative effect of high-risk alleles on PC. In this analysis, we included all four SNPs selected (rs4054823, rs4149117, rs12422149, and rs2735839). The previous analysis revealed that only one SNP showed an independently significant association with PC (rs4054823). The proportions of PC and BPH cases were grouped according to the number of risk alleles that they carried (Figure 1). Generally, the percentage of patients from both groups increased within the subgroups with more than one risk allele; the highest percentage was registered for patients who were carriers of six risk alleles (31% of PC and 32% of BPH). A significant difference was observed between PC and BPH patients when analyzing the distribution of risk alleles within the two study groups; the carriers of eight risk alleles within the PC group were significantly more frequent than the ones within the BPH group (*p* = 0.0023). 

The risk for PC increased significantly with the number of risk alleles (Figure 2). The total risk of PC given by the combination of these four genetic variants was increased to 52%. Therefore, individuals carrying all eight risk alleles of the four SNPs tested had a 1.5-fold increased risk for developing PC compared with the BPH group (OR = 1.49, 95%CI: 1.11–2.10, *p* = 0.0058).

The combined effect of the four selected SNPs was analyzed using a logistic regression model (Table 4). The analysis revealed a stronger cumulative association with PC (*p* = 0.001). The cumulative OR associated with these four regions was 1.42-fold among those subjects who were carriers of at least one risk allele copy at each specific region, compared with subjects without any of these factors.

## 4. Discussion

Due to intensive research in recent years, a large volume of data concerning the association of SNPs with the overall risk for PC is available within the published literature. In genome-wide studies, multiple chromosomal regions have been associated with risk of PC onset and/or progression [14,15,16,17,18]. In this study, our main objective was to examine the cumulative effect of four common polymorphisms on the overall risk of PC. First, we analyzed the association of each of the four SNPs with the PC progression. As a result, we found that SNP rs4054823 shows a strong evidence of PC association (*p* < 0.001; OR = 2.55; 95%CI: 1.44–4.52), whereas rs4149117 shows a weak association (*p* = 0.845; OR = 1.07; 95%CI: 0.50–2.32). In addition, for the other two SNPs on SLCO1B3 and SLCO2B1 genes, the results did not show a statistically significant association with the overall risk of PC (*p* > 0.05). 

Among the tested SNPs in our study, the strongest association with PC was observed for the rs4054823 variant, as the presence of both risk alleles increased the odds of PC 5-fold. Similar findings were reported in a previous study on a large cohort of patients with PC, where the results showed a significant association between rs4054823 polymorphism and the PC progression [12]. In addition, the rs4054823 variant was found to be associated with PC high-grade tumor in a study by Xu et al., where it was concluded that the TT genotype could be also an indicator for aggressive PC at a stage when the disease would potentially be treatable [11]. Furthermore, the results from a previous study found that the polymorphism located on chromosome 17p12 was implicated in the PC progression and was also considered a potential risk factor for PC biochemical recurrence after RP [13].

With regard to rs12422149 and rs4149117, our findings were in contrast with the results published by Cho et al. in a minireview of the literature, where they concluded that genetic variants within the *SLCO1B3* and *SLCO2B1* genes were associated with prostate tumor development and progression [19]. In addition, the results of another study showed that carriers of the variant alleles *SLCO2B1* SNP rs12422149 or *SLCO1B3* SNP rs4149117 had an increased risk of PC-specific mortality [9]. 

Concerning the *KLK2-KLK3* intragenic variant, in our analysis, we did not find a significant association between rs2735839 and PC risk, but we observed a significant association of AG heterozygous genotype and BPH (*p* = 0.042). Our results are in line with those reported by Parikh et al., where the rs2735839 polymorphism did not show a significant association with the PC risk, but they found an association between this marker and the nonaggressive disease [20]. Furthermore, the results of a Swedish population-based case–control study, comprising 1,419 prostate cancer cases and 736 controls, showed no significant association between SNPs in *KLK* genes and PC risk [21]. Similarly, Ahn et al., in a study on a large series of cases and controls, concluded that SNPs in the *KLK3* region were not strongly associated with PC [22]. In contrast to these findings, based on the results of a case–control study, the rs2735839 polymorphism was considered to be a potential risk factor for increased levels of PSA in patients with PC, and, therefore, to be associated with the risk of PC disease [23]. 

Although each of these chromosomal regions was associated with PC risk, in our study, we observed that they have a strong cumulative association with the disease, with a 42% increased risk for PC in carriers of at least one risk allele at each of the four SNPs compared to the subjects without any of the risk variants. It is interesting to see that only one individual SNP was significantly associated with PC and not with BPH, but when analyzing the cumulative effect, we found that all four SNPs tested were significantly increasing the risk for PC (*p* = 0.001). 

Furthermore, we confirmed a strong cumulative effect of genetic variants located on four different chromosomes, namely 11q13, 12p12, 17p12, and 19q13, with PC overall risk. In a similar study by Zheng et al. concerning the cumulative association of different genetic variants with PC, the authors concluded that, although each of the tested SNPs was moderately associated with PC, they had a strong cumulative association with the disease [5]. Using a different approach, we showed a statistically significant association of the sum of all four pre-selected polymorphisms with PC risk. In contrast, another study showed that the vast majority of PC risk-associated SNPs are not associated with aggressiveness and clinicopathologic variables of PC disease, and so, they have minimal utility in predicting the risk for a more aggressive form of PC [24].

## 5. Conclusions

To our knowledge, this study is the first to evaluate the cumulative association between the four susceptibility loci (rs12422149, rs4149117, rs4054823, and rs2735839) located on chromosomes 11q13, 12p12, 17p12, and 19q13, respectively, and the PC overall risk. The data used for the analysis were collected from a homogeneous group of Romanian patients in a clinical setting. Although the present study provides new insights regarding the genetic susceptibility for PC onset and progression, it has some limitations. Due to the small number of participants in this study, the results reported here have limited statistical power in the association analysis; therefore, larger studies are required to verify these findings. It is also worth mentioning that the selected SNPs are based on previous published studies, and, thus, may subject to publication and selection bias. These previous studies did not also include individuals diagnosed with BPH. 

In conclusion, our study provides further evidence regarding the cumulative effect of genetic risk factors on PC overall risk. Our results revealed that the risk of PC was increased significantly among those subjects who were carriers of all eight risk alleles for these selected SNPs. Future studies are needed to investigate the potential risk factors underlying the tumorigenesis mechanism of PC.

Our results should encourage further research in order to explain the mechanism of how these genetic variants interact and how they contribute to PC development and progression. Moreover, it is critical to continue the search for markers that identify PC patients at risk for aggressive disease, as they are more likely to benefit from earlier diagnosis and appropriate treatment.

## Figures and Tables

**Figure 1 biomedicines-10-02733-f001:**
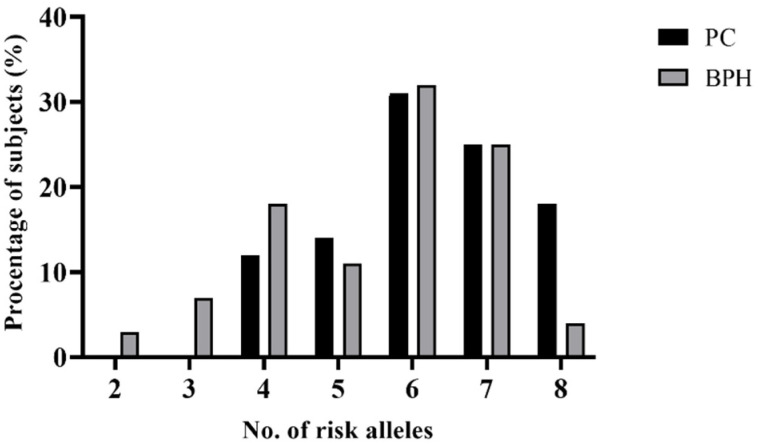
Distributions of risk alleles among patients with PC and BPH.

**Figure 2 biomedicines-10-02733-f002:**
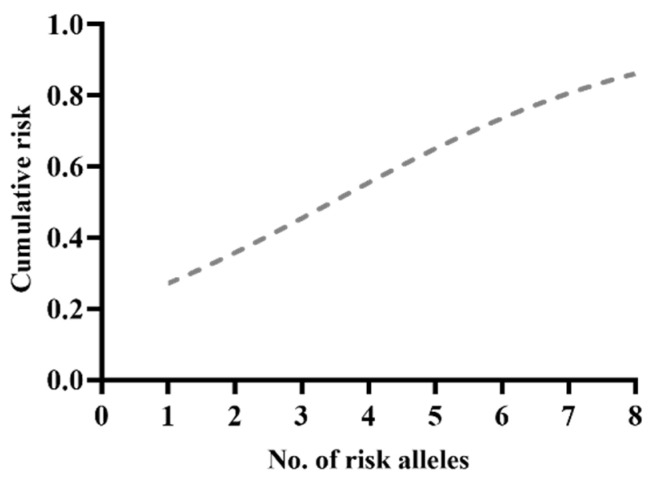
Risk alleles distribution within the PC group.

**Table 1 biomedicines-10-02733-t001:** Clinical characteristics of all participants in the study.

	PC Group (*n* = 50)	BPH Group (*n* = 28)
**Age at diagnosis, years**	65.4 ± 5.2 *	70.5 ± 7.2 *
**˂70**	42	15
**≥70**	8	13
**PSA at diagnosis (ng/mL)**	15.0 ± 5.0 *	11.0 ± 5.0 *
**˂10**	13	14
**≥10**	37	14
**Primary tumor stage (%)**		N/A
**Early**	2 (4)	
**Advanced**	48 (96)	
**Lymph nodes (%)**		N/A
**N0**	31 (62)	
**N1**	19 (38)	
**Gleason score (%)**		N/A
**˂8**	24 (48)	
**≥8**	26 (52)	
**PSM (%)**		
**Yes**	4 (8)	N/A
**No**	46 (92)	
**BCR (%)**		
**Yes**	28 (56)	N/A
**No**	22 (44)	

* Statistics were calculated as mean ± standard deviation; *n* = number of samples.

**Table 2 biomedicines-10-02733-t002:** Genotype frequency of the analyzed SNPs within the PC and BPH groups.

	PC, %	BPH, %	RR, 95% CI	OR, 95% CI	*p* Value
**rs2735839 genotypes**					
**AA**	8	4	1	1	
**AG**	20	39	0.78, 0.61–1.01	0.25, 0.06–0.95	0.042 *
**GG**	72	57	0.96, 0.87–1.06	0.63, 0.18–2.20	0.471
**AG + GG**	92	96	0.95, 0.89–1.02	0.47, 0.13–1.64	0.242
**rs4054823 genotypes**					
**CC**	14	43	1	1	
**CT**	44	32	1.77, 1.31–2.39	4.22, 1.98–8.99	0.002 *
**TT**	42	25	2.04, 1.44–2.88	5.16, 2.36–11.2	<0.001 *
**CT + TT**	86	57	1.50, 1.25–1.82	4.63, 2.32–9.23	<0.001 *
**rs12422149 genotypes**					
**AA**	4	3	1	1	
**AG**	22	36	0.91, 0.76–1.10	0.45, 0.09–2.24	0.335
**GG**	74	61	0.99, 0.92–1.07	0.90, 0.19–4.22	0.904
**AG + GG**	96	97	0.98, 0.93–1.04	0.74, 0.16–3.40	0.701
**rs4149117 genotypes**					
**GG**	4	3	1	1	
**TG**	22	25	0.94, 0.77–1.16	0.66, 0.13–3.27	0.611
**TT**	74	72	0.98, 0.92–1.06	0.77, 0.16–3.56	0.739
**TG + TT**	96	97	0.98, 0.93–1.04	0.74, 0.16–3.40	0.701

* *p* < 0.05 considered to be significant.

**Table 3 biomedicines-10-02733-t003:** Association between all four SNPs tested and PC.

SNP	Allele ^1^	RAF ^2^		OR (95%CI) ^3^	*p* Values
		PC	BPH		
**rs2735839** **(19q13)**	A/**G**	0.82	0.77	1.36 (0.68–2.71)	0.382
**rs4054823** **(17p12)**	C/**T**	0.64	0.41	2.55 (1.44–4.52)	<0.001 *
**rs12422149 (*SLCO2B1*)**	A/**G**	0.85	0.79	1.50 (0.72–3.12)	0.271
**rs4149117 (*SLCO1B3*)**	**T**/G	0.85	0.84	1.07 (0.50–2.32)	0.845

^1^ Minor allele/major allele. The risk alleles are in bold. ^2^ Risk allele frequency (RAF). ^3^ Calculated odds ratio (OR) with 95% confidence interval (CI). * *p* < 0.05 considered to be significant.

**Table 4 biomedicines-10-02733-t004:** Cumulative association of SNPs with prostate cancer.

Region/Gene		17p12	SLCO2B1	SLCO1B3	19q13	*p* Values
**SNPs**		rs4054823	rs12422149	rs4149117	rs2735839	
**Risk allele**		T	G	T	G	
**RAF ^1^ (%)**		Copies of risk alleles ^2^				
**PC**	**BPH**					
**0.04**	0.03	1	2	1	1	0.921
**0.02**	0.04	1	2	2	1	0.782
**0.02**	0.07	2	2	2	1	0.965
**0.04**	0.07	1	2	1	2	0.865
**0.02**	0.07	2	2	1	2	0.116
**0.08**	0.11	1	1	2	2	0.304
**0.16**	0.07	1	2	2	2	0.012 *
**0.20**	0.04	2	2	2	2	0.001 *

^1^ Risk allele frequency (RAF) within the two groups. ^2^ Only carriers of at least one risk allele for all SNPs from both groups were included in the analysis. * *p* < 0.05 considered to be significant.

## Data Availability

All data generated or analyzed during this study are available from the corresponding author on reasonable request.

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
