# Peer review of "Cumulative Effect Assessment of Common Genetic Variants on Prostate Cancer: Preliminary Studies"

_biomedicines, 2022, doi:10.3390/biomedicines10112733_

Round 1

Reviewer 1 Report

In this report Pavel et al characterize the effects of four known SNPs on prostate cancer of Romanian patients. Although, working with a smaller sample, the authors conclusively show that one of the SNPs associate with prostate cancer in these patients. This is unlike other findings reported in American patients where four different SNPs appear to associate with prostate cancer. This report adds to the previous paper from this group (PMID: 35296725). The report is significant and warrants publication because it highlights how various geographic regions are characterized by different risk factors. In this reviewer’s opinion, this study is significant, and I have no major comments.

Minor comments

1.       In the abstract, please spell out the entire names of the acronyms GWAS, PC, OR, BPH and SNP. E.g: Genome wide association studies. This is important because many will skim the abstract and if they don’t know what the acronyms are, they may not proceed to reading your paper.

2.       Lines 147, 156: Please correct the reference error.

Author Response

We would like to thank you for your careful and thorough reading of this manuscript and for the thoughtful comments and constructive suggestions, which help to improve the quality of this manuscript.

Reviewer 2 Report

As prostate cancer is the second most frequent in the world the article entitled: Cumulative effect assessment of common genetic variants on  prostate cancer has the potential impact on the scientific community in the field. The analytical tchicks used for descriptive studies are well-selected and common use. The results of SNP analysis made by TaqMan assays elucidated four critical mutations in the four genes, transversion and transition. The appearing all the mutations together increases the risk of PC by up to 50%! Due to that the obtained data can be redound to new test development. The article is well written and ridable with a small number of typographic errors. However, due to the low number of patients in clinical trials, I strongly recommend the change in the title to Preliminary studies .... Secondly, the number of the Ethical commission is highly demanded for such as studies not only mention that the agreement has been achieved.

In conclusion, the article is the interesting and worth publication, however, without an Ethical commission agreement, it cannot be published.

Author Response

The authors have carefully considered the comments and tried our best to address every one of them.

Round 2

Reviewer 2 Report

In the present form the article is suitable for publication.